# Complicated *Streptococcus agalactiae* Sepsis with/without Meningitis in Young Infants and Newborns: The Clinical and Molecular Characteristics and Outcomes

**DOI:** 10.3390/microorganisms9102094

**Published:** 2021-10-03

**Authors:** Chih Lin, Shih-Ming Chu, Hsiao-Chin Wang, Peng-Hong Yang, Hsuan-Rong Huang, Ming-Chou Chiang, Ren-Huei Fu, Ming-Horng Tsai, Jen-Fu Hsu

**Affiliations:** 1Division of Pediatric Neonatology, Department of Pediatrics, Chang Gung Memorial Hospital, Keelung 204, Taiwan; annielin85@gmail.com (C.L.); kz6479@cgmh.org.tw (S.-M.C.); cyndi0805@yahoo.com.tw (H.-C.W.); qbonbon@gmail.com (H.-R.H.); cmc123@cgmh.org.tw (M.-C.C.); rkenny@cgmh.org.tw (R.-H.F.); 2College of Medicine, Chang Gung University, Taoyuan 333, Taiwan; ph6619@cgmh.org.tw; 3Division of Neonatology, Department of Pediatrics, Chang Gung Memorial Hospital, Chiayi 614, Taiwan; 4Division of Neonatology and Pediatric Hematology/Oncology, Department of Pediatrics, Chang Gung Memorial Hospital, Yunlin 638, Taiwan; 5Division of Neonatology, Department of Pediatrics, Linkou Chang Gung Memorial Hospital, Taoyuan 333, Taiwan

**Keywords:** group B streptococcus, bacterial meningitis, neurological sequelae, late-onset sepsis, intrapartum antibiotic prophylaxis

## Abstract

Background: *Streptococcus agalactiae* (also known as group B streptococcus, GBS) is associated with high mortality and morbidity rates in infants, especially those with complicated GBS sepsis, defined as those with meningitis, severe sepsis and/or septic shock. We aimed to characterize the clinical and molecular characteristics and risk factors for adverse outcomes of neonates with invasive GBS diseases. Methods: From 2003 to 2020, all neonates with invasive GBS diseases who were hospitalized in a tertiary-level neonatal intensive care unit (NICU) were enrolled. The GBS isolates underwent serotyping, multilocus sequence typing (MLST) and antibiotic susceptibility testing. We compared cases of complicated GBS sepsis with uncomplicated GBS bacteremia. Results: During the study period, a total of 188 neonates (aged less than 6 months old) with invasive GBS diseases were identified and enrolled. Among them, 119 (63.3%) had uncomplicated GBS bacteremia and 69 (36.7%) neonates had complicated GBS sepsis, including meningitis (25.5%, *n* = 48) and severe sepsis or septic shock. Among neonates with complicated GBS sepsis, 45 (65.2%) had neurological complications, and 21 (42.0%) of 50 survivors had neurological sequelae at discharge. The overall final mortality rate was 10.1% (19 neonates died). Type III/ST-17 GBS isolates accounted for 56.5% of all complicated GBS sepsis and 68.8% of all GBS meningitis, but this strain was not significantly associated with worse outcomes. The antimicrobial resistance rate among the invasive GBS isolates was obviously increasing in the past two decades. After multivariate logistic regression analysis, neonates with thrombocytopenia and respiratory failure were independently associated with final adverse outcomes. Conclusions: a total of 36.7% of all neonatal invasive GBS diseases were associated with complicated sepsis with/without meningitis. Given the high mortality and morbidity rates in neonates with complicated GBS sepsis, further studies for early identification of specific strains, risk factors or genetic mechanisms that will cause complicated GBS sepsis are urgently needed in the future.

## 1. Introduction

*Streptococcus agalactiae* (also known as group B streptococcus, GBS) is the most important and leading pathogen that causes bacterial meningitis in neonates [1,2]. Recent studies have found that the incidence of GBS early-onset disease (EOD, disease occurring within the first week of life) decreases, but GBS late-onset disease (LOD, disease occurring 8–90 days of age) remains unchanged in the era of routine GBS screening and maternal intrapartum prophylaxis [3,4]. The incidence of invasive GBS in young infants varies greatly worldwide, from 0.55 per 1,000 live births in Asia to 1.21 per 1000 live births in Africa [3,4,5,6]. Our previous studies found that nearly one-third of neonates with invasive GBS disease had meningitis, and 31.8% of them had infectious complications [7,8,9]. Neonatal GBS meningitis not only causes a high mortality rate but also results in neurological complications and long-term neurodevelopmental sequelae [10,11,12,13].

Most studies regarding invasive GBS diseases or meningitis focus on the epidemiology, molecular characteristics, antifungal susceptibility and outcomes [3,4,5,6,7,8]. Little is known about the clinical and microbiological characteristics of complicated GBS sepsis, including cases of meningitis and severe sepsis with/without septic shock [13,14]. Furthermore, the neurological sequelae and complications after GBS meningitis are supposed to be different from those caused by non-GBS bacterial pathogens [1,14,15,16]. Understanding the correlation between GBS molecular epidemiology and clinical features may prompt optimized therapeutic and preventive strategies [14,15,16]. In this study, we describe the clinical features, complications, outcomes and bacterial serotype distribution of complicated GBS sepsis and GBS meningitis from a tertiary medical center in Taiwan.

## 2. Methods

### 2.1. Study Design and GBS Isolates

Between January 2003 and December 2020, all neonates aged less than six months old with invasive GBS diseases were enrolled and their data were retrieved from the neonatal intensive care unit (NICU) database of Chang Gung Memorial Hospital (CGMH), a tertiary level medical center in North Taiwan. We reviewed the electronic chart records for patients’ demographics, clinical characteristics, hospital courses, treatment outcomes, and long-term follow-up. All GBS isolates were obtained from the bacterial library of CGMH’s central laboratory. We did not enroll GBS isolates from sputum or bronchoalveolar lavage fluid. This study was approved by the Institutional Review Board of CGMH (IRB No. 104-6818B), and a waiver of informed consent for anonymous data collection was approved.

### 2.2. Definition

Invasive GBS disease was defined as GBS infection with GBS strains isolated from a sterile site, including blood, urine, cerebrospinal fluid (CSF), soft tissues (necrotic tissues, abscesses, or cellulitis) and pleural or peritoneal fluids. Meningitis was defined by the World Health Organization as the presence of clinical signs of possible serious bacterial infection [17,18] and CSF culture positive for bacterial pathogens or blood culture/polymerase chain reaction (PCR)/latex agglutination positive for bacterial pathogens with a CSF leukocyte count > 20 × 106/L. Episodes reported by physicians with negative CSF cultures were also included if CSF results showed at least one individual marker of bacterial meningitis (defined as a glucose level of less than 34 mg/dL [1.9 mlol/L], a ratio of CSF glucose to blood glucose of less than 0.23, a protein level of more than 220 mg/dL, or a leukocyte count of more than 2000/μL) [19] and the clinical presentation was compatible with bacterial meningitis. For severe sepsis and uncomplicated bacteremia, we applied the definitions of the Centers for Disease Control and Prevention [20]. The presence of neurological complications and long-term neurological sequelae of these patients were evaluated based on the definitions in previous studies [10,21].

### 2.3. Capsular Serotyping and MLST

The multiplex PCR assay was used to analyze the capsular serotypes for identifying types Ia to IX of GBS isolates. The DNA isolation method and the PCR assay, amplifying and sequencing seven housekeeping genes (*adhP*, *atr*, *glcK*, *glnA*, *pheS*, *sdhA* and *tkt*), were based on standard protocol and described in our previous publication [22]. Multilocus sequence typing (MLST) was then conducted based on the standard procedure as described in our previous studies [23]. After PCR, the sequence type (ST) was assigned based on the allelic profile of each fragment and determined via the Streptococcus agalactiae MLST database (http://pubmist.org/sagalactiae, acessed on 31 August 2021). All GBS isolates can be clustered into several major clonal complexes (CCs) based on the goeBURST program [23].

### 2.4. Antimicrobial Susceptibility Testing

Antimicrobial susceptibility testing was performed by the agar diffusion method as described in previous studies [24]. The double-disk diffusion test was applied to identify inducible clindamycin resistance. All GBS isolates were rated for susceptibility to seven antibiotics, including erythromycin, penicillin, clindamycin, vancomycin, ampicillin, cefotaxime and teicoplanin according to the guidelines of the Clinical and Laboratory Standards Institute for the microdilution minimum inhibitory concentration (MIC) method [25]. 

### 2.5. Statistical Analysis

We categorized neonatal invasive diseases into two subgroups: uncomplicated GBS bacteremia and complicated GBS sepsis with/without meningitis. Categorical and continuous variables were expressed as proportions and the median (interquartile, IQR), respectively. Categorical variables were compared using the χ2 test or Fisher’s exact test; odds ratios (ORs) and 95% confidence intervals (CIs) were calculated. Continuous variables were compared using the Mann–Whitney U-test and the t-test, depending on the distributions. The trend of proportions of the categorical variables among the subgroups was analyzed by the Cochran–Armitage trend test. *p* values of <0.05 were considered statistically significant.

We also aimed to identify the independent risk factor for final adverse outcomes of neonates with invasive GBS diseases, which included cases of final in-hospital mortality and neonates with neurological sequelae at discharge. Associations between patients’ demographic, clinical and molecular characteristics, and laboratory results were tested in univariate analyses, and odds ratios (ORs) were used to quantify the strength of association. Covariates presumed to be associated with final adverse outcomes based on previous studies and those associated with mortality at *p* < 0.1 were subsequently entered into multivariable logistic regression models. All statistical analyses were analyzed using SPSS version 23 (IBM SPSS Statistics).

## 3. Results

During the study period, a total of 188 cases of invasive GBS diseases in neonates were identified and analyzed. There were a total of 44 EODs (23.4%) and 144 LODs (76.6%). Only 11 cases were more than 90 days old. The median (interquartile range [IQR]) gestational age and birth body weight of the cohort were 38.0 (36.3–39.0) weeks and 2885 (2550–3235) g, respectively. In the cohort, 141 (75.0%) were term-born (GA ≥ 37 weeks) neonates, only 10 (5.3%) were extremely preterm (GA ≤ 28 weeks) and 15 (8.0%) of them were very low birth weight infants (BBW < 1500g). The median (IQR) onset of GBS invasive diseases were 25.5 (9.3–53.8) days old.

### 3.1. Complicated GBS Sepsis versus Uncomplicated GBS Bacteremia

Neonates with GBS meningitis, GBS severe sepsis and/or septic shock were categorized as complicated GBS sepsis. Otherwise, they were categorized as uncomplicated GBS bacteremia. In our cohort, 36.7% (*n* = 69) were cases of complicated GBS sepsis and 63.3% (*n* = 119) were uncomplicated GBS bacteremia (Table 1). A total of 48 cases (25.5%) of all GBS invasive diseases had meningitis. We found that neonates with complicated GBS sepsis had significantly more severe clinical manifestations and abnormal laboratory findings than those with uncomplicated GBS bacteremia (Table 1). All these neonates were treated with ampicillin since none of these isolates were resistant to ampicillin. In addition, GBS EOS cases were significantly more likely to be complicated GBS sepsis than GBS LOS cases (59.1% vs. 29.9%, *p* < 0.001). The overall final mortality rate was 10.1% (19 patients died). In addition, 23.9% (*n* = 45) had neurological complications during hospitalization (Table 2), and 12.4% (21/169) of the survivors had neurological sequelae at discharge.

All the GBS invasive isolates underwent serotype and MLST analyses. The MLST analyses revealed seven serotypes, with serotype III GBS predominant (67.0%), and followed by serotype Ia (16.5%) and eleven different STs: these STs could be clustered into nine CCs (Figure 1). ST17 was the most frequent ST and accounted for 60.1% of all isolates, followed by ST1 and ST12 (both 7.4%). All STs can be found in the GBS MLST database (http://pubmist.org/sagalactiae, accessed on 31 August 2021). A high correlation between STs and serotypes of the 188 GBS invasive isolates was noted and is presented in this study; this is presented in Appendix A. The notable correlations between serotype III and ST17, serotype Ib and ST12, and serotype VI and ST1 were evident, as ST17, ST12 and ST1 accounted for 89.7% (*n* = 113), 92.9% (*n* = 13) and 100% (*n* = 5) of all serotype III, serotype Ib and serotype VI isolates, respectively.

We found that neonates with complicated GBS sepsis had a high proportion of neurological complications (*n* = 45, 65.2%) and the survivors were likely to have neurological sequelae (*n* = 21, 42.0%) at discharge (Table 2). The molecular characteristics of these invasive GBS isolates were correlated with clinical manifestations (Figure 1). Notably, type III/ST-17 GBS isolates accounted for 56.5% of all complicated GBS sepsis and 68.8% of all cases of GBS meningitis. However, serotype II/ST-1 and serotype Ib/ST12 GBS isolates had the highest percentage of causing complicated GBS sepsis (3/4, 75.0% and 9/14, 64.3%, respectively) in neonates compared with serotype III GBS isolates (both *p* < 0.001). During the study period, the serotype III/CC17 GBS strain became predominant since 2006, and accounted for more than 70–75% of all neonatal invasive strains since 2012 (Figure 2). In addition, there was a significant trend of decreased percentages of complicated GBS sepsis and meningitis in neonates (Figure 3), and the mortality rate has been decreased from 17.8% during 2003–2011 to 5.2% during 2012–2020.

A significantly higher rate of unfavorable outcomes was observed in patients with neurological complications compared with those without any neurological complications (73.3% vs. 4.9%, *p* < 0.001). We aimed to find significant predictors of final unfavorable outcomes, including final mortality (*n* = 19) and neonates with neurological sequelae at discharge (*n* = 21). We found that initial septic shock, respiratory failure requiring intubation, thrombocytopenia, leukopenia, preterm neonates, EOD and type Ib/CC12 GBS isolates were significantly associated with a higher risk of final unfavorable outcomes (Table 3). Although the type III/CC-17 GBS isolates accounted for more than half of complicated GBS sepsis and GBS meningitis, they did not significantly lead to a worse outcome. After multivariate logistic regression, neonates with respiratory failure requiring intubation (adjusted odds ratio [aOR]: 14.97, 95% confidence interval [CI]: 3.31–67.79, *p* < 0.001) and thrombocytopenia (aOR: 6.19; 95% CI: 1.88–20.33, *p* = 0.003) were the independent predictors for final unfavorable outcomes.

### 3.2. Antimicrobial Susceptibility Results

All the invasive GBS isolates were susceptible to penicillin, ampicillin, and vancomycin. None of these strains showed cefotaxime resistance. The overall resistance to erythromycin and clindamycin was 72.3% and 70.7%, respectively. Most of the GBS isolates that were erythromycin resistant were also clindamycin resistant (*n* = 129, 94.9%). As we observed the antimicrobial resistance profiles by different serotypes and STs, a significantly higher antibiotic resistance rate (to erythromycin and clindamycin) was noted in serotype Ib (100%), type V (85.7%), and type III GBS isolates (77.1–82.2%). The antibiotic resistance rate to erythromycin and clindamycin was especially high in ST12 (100%) and ST17 (89.3%), respectively. There was no significant difference in the percentage of antimicrobial GBS isolates between subgroups of complicated GBS sepsis and uncomplicated GBS bacteremia. During the past two decades, an increasing trend of serotype III/ST17 GBS accounted for the increasing trend of the antibiotic resistance rate (Figure 4).

## 4. Discussion

To our knowledge, this is the first study to investigate the issue of complicated GBS sepsis in neonates [14], including GBS meningitis and GBS severe sepsis with/without septic shock. In our cohort, 36.7% of GBS invasive diseases in neonates were associated with complicated courses, a high rate of neurological complications and sequelae, and a high risk of mortality. Although serotype III/ST-17 GBS isolates accounted for more than half of all neonates with complicated GBS sepsis and meningitis, serotype Ib/CC12 GBS isolates were significantly more likely to cause complicated GBS sepsis (Figure 1). Similar to recent studies, the increasing trend of serotype III/CC17 GBS predominance and antibiotic resistance rate among neonatal invasive diseases were noted in the past two decades [3,4,5,26,27,28]. After multivariate logistic regression analysis, respiratory failure requiring intubation and thrombocytopenia were independently associated with final adverse outcomes.

Routine GBS screening for pregnant women and intrapartum antibiotic prophylaxis (IAP) have been initiated since 2004 in our institute. Therefore, serotype III/ST-17 GBS isolates have been predominant since 2006 and account for more than two-thirds of all neonatal GBS invasive diseases since 2012. Although cases of neonatal GBS sepsis have significantly increased since 2012, which may be due to the expansion of referrals from other local hospitals in Taiwan, the case numbers and incidence rate of neonatal GBS sepsis have declined since 2018, which can be explained by the universal GBS screening and IAP in all local hospitals in Taiwan. Recent studies have found an increasing incidence of LOD [3,26,27,28], but most of them only enrolled cases of neonatal invasive GBS diseases until 2014 or 2015 [2,3,4,5,26,27,28,29,30]. Some updated information had data on GBS surveillance until 2017 or 2018 [31,32,33], while this report is one of the very few studies that enrolled neonatal invasive GBS diseases until 2020. We can conclude that the significant effects of general and universal GBS screening and IAP since 2012 in Taiwan have led to decreased invasive GBS diseases in neonates since 2018, in both EOD and LOD.

In our cohort, only neonates with complicated GBS sepsis would have adverse outcomes and a recent study documented that shortened IV antibiotic courses can be prescribed [14]. An obviously decreased GBS sepsis-attributable mortality rate has been reported in recent decades, from more than 20% in the 1980s to approximately 10.7% in recent studies [3,4,5,26,27]. The overall mortality rate in our cohort was compatible with recent studies [3,4,5,26,27,28,29,30,31,32,33] and a reduction in mortality between 2012 and 2020 was observed in our cohort. As reported in recent studies, multiple factors may account for the reduced mortality rate, including widespread implementation of antibiotics in at-risk mothers and babies and advances in managing acutely ill neonates [33,34]. Based on our data, neonatal GBS EOD had a significantly higher rate of complicated sepsis and a higher rate of final mortality than GBS LOD. Therefore, it was reasonable that IAP reduced the occurrence of GBS EOD, which subsequently led to improved outcomes in our institute. We suspected that the occurrence of complicated GBS sepsis is associated with both host factors and factors of highly pathogenic GBS strains [35,36] because some healthy late preterm or term-born neonates without underlying chronic comorbidities had a very high illness severity. Therefore, further studies for early identification of specific strains, risk factors or genetic mechanisms that will cause complicated GBS sepsis are urgently needed in the future.

We found that some neonates with complicated GBS sepsis but without evidence of meningitis or ventriculitis had neurological complications or sequelae. In neonates with complicated GBS sepsis, nearly two-thirds developed at least one neurological complication in the acute or subacute phase. The bacteremia-associated neurological complications may result from either direct bacterial invasion, sepsis-associated encephalopathy [10,37,38], or both. Our previous study concluded that the presence of septic shock and GBS infections were independently associated with a 5.9- and a 8.9-fold increased risk of developing neurological complications in neonatal bacteremia with meningitis, respectively [10,38]. Currently the odds ratio has been decreased since 2012, which may be due to the improved neonatal care and advanced knowledge of severe GBS sepsis.

Additionally, in our neonates with complicated GBS sepsis, a significant proportion of neurological complications, including hydrocephalus, ventriculomegaly, brain infarction and encephalomalacia were detected approximately several weeks after bacteremia onset. Their delayed diagnoses may be explained by subtle or nonspecific symptoms. These intracranial lesions are supposed to result from subacute neurological damage, which is different from the focal infectious complications, such as subdural empyema or abscess, which probably are due to direct extension or persistence of bacterial invasion and are easily identified in the acute stage [10,38,39]. Neonates with hydrocephalus, encephalomalacia or infarction are at high risk of developing long-term neurological sequelae and neurodevelopmental delay [40,41]. Therefore, more aggressive strategies in neonates with complicated GBS sepsis and early identification of neurological complications and the intracranial lesions using neuroimaging studies are needed.

The increasing antimicrobial resistance rate in our invasive GBS isolates was due to the emerging predominance of serotype III/CC17 GBS isolates, which had a significantly high antibiotic resistance rate. It is worth noting that serotype Ib/ST12 GBS isolates had the highest antibiotic resistance rate and were likely to cause severe neonatal diseases [42]. Serotype Ib/ST10 or ST12 GBS strains are not common isolates that cause neonatal invasive diseases, but they were reported to emerge in Northern and Eastern China [42,43] or Taiwan. Therefore, continued surveillance of GBS isolates and monitoring of type Ib GBS infections is warranted, given that they are potentially associated with the emergence of multidrug-resistant isolates, likely to cause severe infections and have clonal expansion [43,44].

There are some limitations in this study. All invasive GBS isolates were from a single center in Taiwan, and we could not control case referrals or transfers from local hospitals. Therefore, we were unable to provide the incidence rate and trend of neonatal invasive diseases during the long study period. Compared with some nationwide surveillance or meta-analyses [2,3,4,31,41], the case number in this study was small. It is difficult to define the clinical importance of some GBS strains, including types III, V and VI. Additionally, only the PCR method was used to investigate the molecular characteristics of our GBS isolates, and we did not detect the expression of genes. Finally, some early mortality cases and some invasive GBS strains more than 10 years ago might have been missed and were not included in this study.

## 5. Conclusions

In conclusion, in contrast with uncomplicated GBS bacteremia, we found that neonates with complicated GBS sepsis are at high risk of mortality, neurological complications, and neurological sequelae. There is an increasing trend of antibiotic resistance to erythromycin and clindamycin in our institute. Although the percentages of severe GBS infection, meningitis and mortality have been decreased in recent years, some pathogenic GBS isolates, including type III/ST-17 and type Ib/ST-12 strains, are emerging and may cause neonatal invasive diseases. Therefore, continuous monitoring of neonatal invasive GBS diseases, especially those with complicated GBS sepsis remains warranted in the future.

## Figures and Tables

**Figure 1 microorganisms-09-02094-f001:**
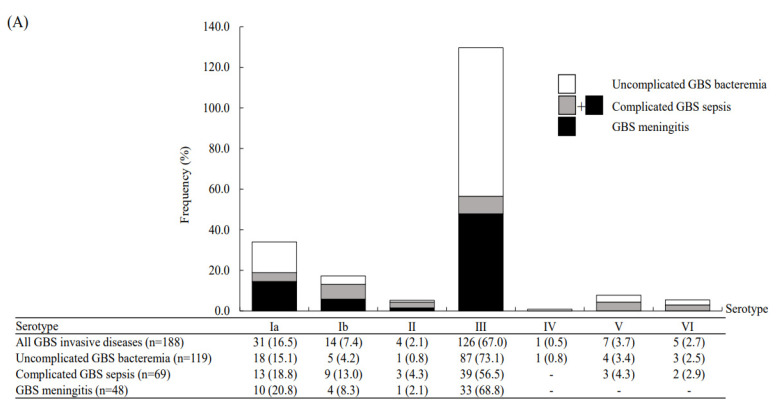
Distribution of all group B streptococcal (GBS) invasive diseases in neonates between 2003 and 2020, stratified by uncomplicated GBS bacteremia, complicated GBS and GBS meningitis: (**A**) The distribution of capsular serotypes; (**B**) the distribution of sequence types (ST); (**C**) the distribution of clonal complex (CC) types.

**Figure 2 microorganisms-09-02094-f002:**
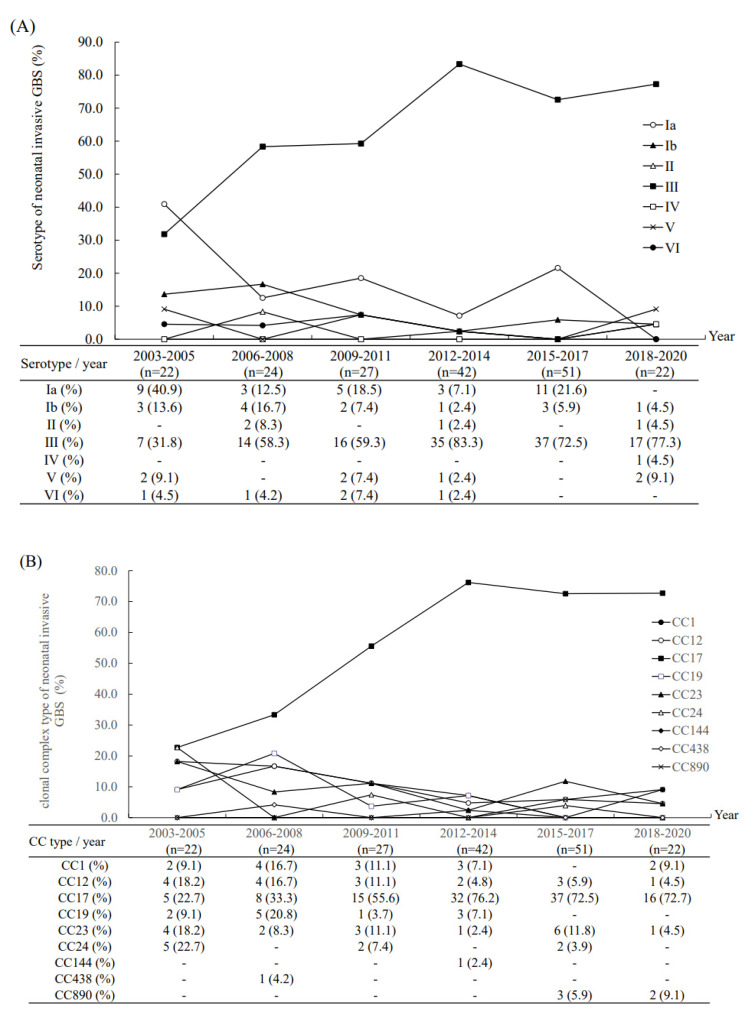
The percentage of all group B streptococcal (GBS) invasive diseases in neonates hospitalized in the neonatal intensive care unit of Chang Gung Memorial Hospital during an 18-year study period: (**A**) The percentage of six serotypes in six time periods; (**B**) the percentage of clonal complexes in six time periods.

**Figure 3 microorganisms-09-02094-f003:**
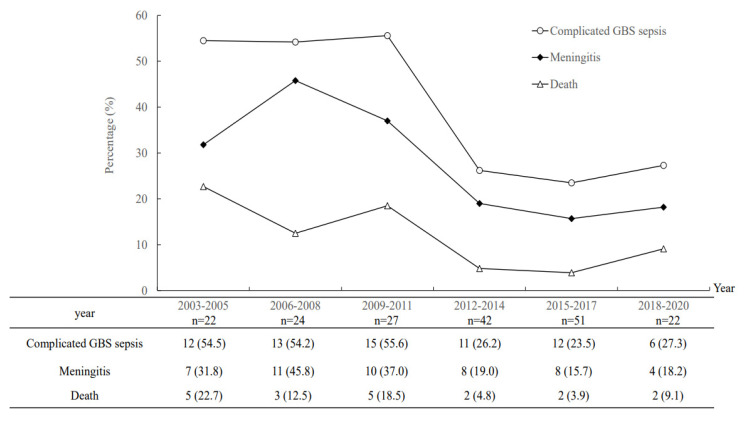
The percentage of complicated group B streptococcal sepsis, GBS meningitis and mortality rate of neonatal invasive GBS diseases from 2003 to 2020.

**Figure 4 microorganisms-09-02094-f004:**
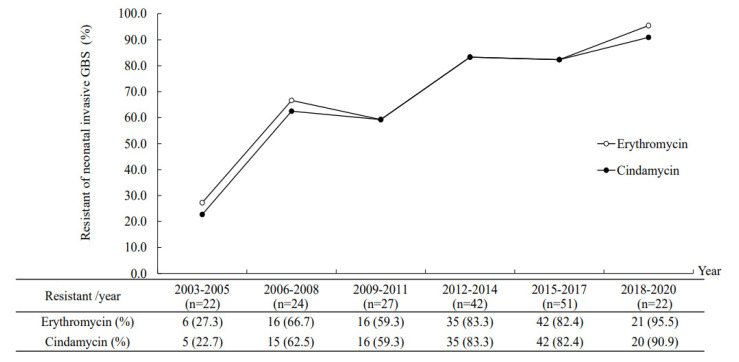
The increasing antimicrobial resistance to erythromycin and clindamycin in 188 invasive group B streptococcal isolates from 2003 to 2020.

**Table 1 microorganisms-09-02094-t001:** Patient demographics of neonatal invasive group B streptococcus (GBS) infections in Chang Gung Memorial Hospital (CGMH).

Demographics	All Cases (Total *n* = 188)	Complicated GBS Infections with/without Meningitis (Total *n* = 69)	Uncomplicated GBS Bacteremia (Total *n* = 119)	*p* Values
Gestational age (week), median (IQR)	38.0 (36.3–39.0)	38.0 (36.0–39.0)	39.0 (37.0–40.0)	0.049
Birth body weight (g), median (IQR)	2885.0 (2551–3235)	2750.0 (2490–3120)	2960.0 (2640–3305)	0.039
Gender, (male/female, *n*/%)	81 (43.1)/107 (56.9)	31 (44.9)/38 (55.1)	50 (42.0)/69 (58.0)	0.761
Birth by NSD/Cesarean section, *n* (%)	130 (69.1)/58 (30.9)	45 (65.2)/24 (34.8)	85 (71.4)/34 (28.6)	0.414
5 min Apgar score < 7, *n* (%)	12 (6.4)	6 (8.7)	6 (5.0)	0.362
Premature rupture of membrane, *n* (%)	35 (18.6)	20 (29.0)	15 (12.6)	0.007
Onset of GBS bacteremia (day), median (IQR)	25.5 (9.3–53.8)	18.0 (2.0–32.5)	30.0 (15.0–56.0)	0.001
Early-onset sepsis (≤7 days), *n* (%)	44 (23.4)	26 (37.7)	18 (15.1)	0.002
Late-onset sepsis (8–90 days), *n* (%)	133 (70.7)	40 (58.0)	93 (78.2)	0.002
Very late-onset sepsis (>90 days), *n* (%)	11 (5.9)	3 (4.3)	8 (6.7)	0.142
Clinical features *, *n* (%)				
Fever (≥38.3 ℃)	153 (81.4)	49 (71.0)	104 (87.4)	0.007
Apnea, bradycardia and/or cyanosis	67 (35.6)	54 (78.3)	13 (10.9)	<0.001
Ventilator requirement				<0.001
Room air or nasal canuala	124 (66.0)	23 (33.3)	101 (84.9)	
Non-invasive ventilator (N-CPAP and N-IMV)	18 (9.6)	3 (4.3)	15 (12.6)	
Intubation	37 (19.7)	34 (49.3)	3 (2.5)	
High-frequency oscillatory ventilator	9 (4.8)	9 (13.0)	0 (0)	
Abdominal distension and/or vomiting	71 (37.8)	40 (58.0)	31 (26.1)	<0.001
Hypoglycemia	22 (11.7)	13 (18.8)	9 (7.6)	0.032
Hypotension	34 (18.1)	33 (47.8)	1 (0.8)	<0.001
Severe sepsis	39 (33.1)	39 (56.5)	0 (0)	<0.001
Disseminated intravascular coagulopathy	13 (6.9)	13 (18.8)	0 (0)	<0.001
Requirement of blood transfusion **	88 (46.8)	38 (55.1)	50 (42.0)	0.096
Laboratory data at onset of GBS bacteremia, *n* (%)				
Leukocytosis (WBC > 20,000/L)	110 (58.5)	35 (50.7)	75 (63.0)	0.125
Leukopenia (WBC < 4000/L)	41 (21.8)	24 (34.8)	17 (14.3)	0.002
Shift to left in WBC (immature > 20%)	23 (12.2)	11 (15.9)	12 (10.1)	0.255
Anemia (hemoglobin level < 11.5 g/dL)	97 (51.6)	38 (55.1)	59 (49.6)	0.545
Thrombocytopenia (platelet < 150,000/μL)	31 (16.5)	24 (34.8)	7 (5.9)	<0.001
Metabolic acidosis	27 (14.4)	23 (33.3)	4 (3.4)	<0.001
Coagulopathy	30 (16.0)	27 (39.1)	3 (2.5)	<0.001
C-reactive protein (mg/dL), median (IQR)	40.7 (10.8–104.0)	83.2 (21.7–167.9)	20.4 (7.0–57.6)	<0.001

* At onset of GBS bacteremia; ** including leukocyte poor red blood cell and/or platelet transfusion. All data are expressed as number (%) or median (IQR). IQR: interquartile range; WBC: white blood cell count; N-CPAP: nasal continuous positive airway pressure; N-IMV: non-invasive mechanical ventilation.

**Table 2 microorganisms-09-02094-t002:** Neurological complications in neonates with complicated group B streptococcal (GBS) sepsis in CGMH, 2003–2020.

Neurological Complications, Sequelae and Death	Neonates with Complicated GBS Sepsis and Meningitis (*n* = 48)	Neonates with Severe Sepsis and/or Septic Shock (*n* = 21)
Any neurological complications	37 (77.8)	8 (38.1)
Seizure	22 (45.8)	5 (23.8)
Subdural effusion	16 (33.3)	1 (4.8)
Increased intracranial pressure	12 (25.0)	7 (33.3)
Ventriculomegaly	17 (35.4)	0 (0)
Hydrocephalus	6 (12.5)	1 (4.8)
Encephalomalacia	6 (12.5)	0 (0)
Subependymal hemorrhage	5 (10.4)	2 (9.5)
Intraventricular hemorrhage	4 (8.3)	4 (19.0)
Ventriculitis	4 (8.3)	0 (0)
Periventricular leukomalacia	1 (2.1)	1 (4.8)
Infarction	5 (10.4)	0 (0)
Subdural empyema or abscess	2 (4.2)	0 (0)
Brain atrophy	1 (2.1)	0 (0)
Discharge with neurological sequelae	17 (35.4)	4 (19.0)
Final in-hospital mortality	7 (14.6)	12 (57.1)

All data are expressed as number (%).

**Table 3 microorganisms-09-02094-t003:** Risk factors for final unfavorable outcomes (death or major neurological sequelae at discharge) by univariate and multivariate analysis.

Parameters	Univariate Analysis		Multivariate Analysis	
	OR (95% CI)	*p* Value	Adjusted OR (95% CI)	*p* Value
Preterm birth (GA < 37 weeks)	2.52 (1.19–5.31)	0.015	0.70 (0.24–2.08)	0.525
Septic shock	20.7 (8.41–50.97)	<0.001	1.40 (0.32–6.11)	0.654
Respiratory failure (requirement of intubation)	24.75 (10.23–59.91)	<0.001	14.97 (3.31–67.79)	<0.001
Type III/CC-17 GBS isolates	0.24 (0.12–0.50)	<0.001		
Type Ib/CC-12 GBS isolates	5.92 (1.92–18.24)	0.002	2.11 (0.45–9.88)	0.344
Early-onset sepsis	3.84 (1.81–8.15)	<0.001	1.70 (0.70–2.42)	0.570
Leukopenia (WBC count < 5000 cells/μL)	2.41 (1.12–5.22)	0.025	0.77 (0.23–2.52)	0.660
Anemia (hemoglobin level < 11.5 mg/dL)	1.35 (0.67–2.74)	0.401		
Thrombocytopenia (platelet count < 150,000/μL)	13.8 (5.65–33.68)	<0.001	6.19 (1.88–20.33)	0.003

GA: gestational age; CSF: cerebrospinal fluid; OR: odds ratio; 95% CI: 95% confidence interval.

## Data Availability

The datasets used/or analyzed during the current study are available from the corresponding author on reasonable request.

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
