# Peer review of "Complicated Streptococcus agalactiae Sepsis with/without Meningitis in Young Infants and Newborns: The Clinical and Molecular Characteristics and Outcomes"

_microorganisms, 2021, doi:10.3390/microorganisms9102094_

Round 1

Reviewer 1 Report

Please remove the short title

Please shorten the abstract to include the essential information, focusing on the results.

„There was an obviously increasing antimicrobial resistance rate among these invasive GBS isolates in the past two decades.” please rephrase

Please adhere to the international guidelines for writing bacterial names! (italics, first mention, subsequent mentions…)

Introduction:

As a starting sentence, please discuss the different groups of streptococci, including GAS, GCS/GGS, enterococci, and non-typable strains (S. suis). Consider using the following references:

https://pubmed.ncbi.nlm.nih.gov/33408489/

https://pubmed.ncbi.nlm.nih.gov/32847011/

Methods:

please separate subsection 2.1. into more independent sections.

Please adhere to the writing of representative genes!

which standard was used for disk diffusion? this is ambiguous, please clarify

Figure 1A,B and C: the tables under the figures and the legend is too little and small to read, this must be corrected and addressed.

Figure 2A and B should be increased to the same size of Figure 3.

I suggest that the authors remove figure 4 or amend it, as this information could have been presented in the part of the text.

Discussion: too many sentences begin with „We found that …” please rephrase.

Please make sure to contextualize your study results in the context of other literature findings.

Author Response

RE: microorganisms-1384247

Complicated Group B Streptococcus sepsis with/without meningitis in young infants and newborns: the clinical and molecular characteristics and outcomes

Dear Editor,

Thank you for your appreciated comments on our manuscript. We had the manuscript revised, all according to the reviewers’ and editor’s suggestions. We underline every change and highlight in red color on the revised manuscript. The replies for the reviewers’ criticisms are as followings. We hope this revised version can be acceptable.

Best regards,

Ming-Horng Tsai

Chief, Division of Neonatology and Pediatric Hematology/Oncology, Department of Pediatrics, Yunlin Chang Gung Memorial Hospital, Taiwan, R.O.C.

Comments from Reviewer No.1:

Reviewer No.1

Please remove the short title

Reply:

      Thank you for your instructive advice. I will remove the short title accordingly, thank you.

Please shorten the abstract to include the essential information, focusing on the results.

Reply:

      Thank you for your instructive advice. I will shorten the abstract to include the essential information, focusing on the results, thank you. The revised abstract has been shortened, thank you.

There was an obviously increasing antimicrobial resistance rate among these invasive GBS isolates in the past two decades.” please rephrase

Reply:

      Thank you for your instructive advice. I rephrase as the following: The antimicrobial resistance rate among the invasive GBS isolates was obviously increasing in the past two decades. Thank you.

Please adhere to the international guidelines for writing bacterial names! (italics, first mention, subsequent mentions…)

Reply:

      Thank you for your instructive advice. I will adhere to the international guidelines for writing bacterial names, including italics, first mention, and subsequent mentions, thank you. I used Streptococcus agalactiae (also known as Group B Streptococcus, GBS) in the abstract and the first paragraph of the introduction, thank you.

Introduction:

As a starting sentence, please discuss the different groups of streptococci, including GAS, GCS/GGS, enterococci, and non-typable strains (S. suis). Consider using the following references:

https://pubmed.ncbi.nlm.nih.gov/33408489/

https://pubmed.ncbi.nlm.nih.gov/32847011/

Reply:

      Thank you for your instructive advice. Initially I tried, later I found it is very difficult. It is very difficult to discuss the different groups of streptococci, including GBS, GCS/GGS, enterococci, and non-typable strains in the manuscript, because we focused on the complicated GBS sepsis with/without meningitis in neonates. In addition, there have been many GBS studies in the literature, and I found none of them mentioned the different groups of streptococci in the introduction section. I am sorry about that. Please forgive me, it is too difficult.

Methods: please separate subsection 2.1. into more independent sections.

Reply:

      Thank you for your instructive advice. I will separate this subsection 2.1. into two independent sections (new subsection 2.1 and 2.2), thank you.

Please adhere to the writing of representative genes!

Reply:

Thank you for your instructive advice.  I will revise them as adhP, atr, glcK, glnA, pheS, sdhA, and tkt) in the revised manuscript, thank you.

Which standard was used for disk diffusion? this is ambiguous, please clarify

Reply:

       Thank you for your instructive advice. We used agar dilution method to determine the MICs of seven antibiotics according to the CLSI criteria. I revise this issue in line 14 of page 3, by the agar diffusion method, thank you.

Figure 1A, B and C: the tables under the figures and the legend is too little and small to read, this must be corrected and addressed.

Reply:

        Thank you for your instructive advice.  I will correct the tables under the figures and the legends. I will enlarge them in the revised manuscript, thank you.

Figure 2A and B should be increased to the same size of Figure 3.

Reply:

Thank you for your instructive advice. I will increase the size of figure 2A and B according, thank you.

I suggest that the authors remove figure 4 or amend it, as this information could have been presented in the part of the text.

Reply:

Thank you for your instructive advice. It is not difficult to have the information of figure 4 in the part of the text. However, in figure 4, we can have all the antimicrobial resistant rates and the numbers of resistant isolates to two different antimicrobials during the six study periods. There will be too many words to present all the data in the text. In addition, a figure can clearly present the increasing trend of antimicrobial resistant rate among these GBS isolates. Therefore, I suggest to keep the figure 4, thank you.

Discussion: too many sentences begin with „We found that …” please rephrase.

Reply:

       Thank you for your instructive advice.  I will remove “We found that…..” as much as I can in the revised manuscript, thank you.

       Please notice that I have deleted or rephrased “we found that” four times in the discussion section, and now only one (in the beginning of the fourth paragraph) is left in the revised manuscript, thank you.

Please make sure to contextualize your study results in the context of other literature findings.

Reply:

       Thank you for your instructive advice. I will make sure to contextualize my study results in the context of other literature findings, thank you.

      Please notice the references no. 31-35 and no. 38,40-43 are good examples. I have contextualized my study results in the context of other literature findings.

Reviewer 2 Report

In this paper, Lin and collaborators compared cases of complicated and uncomplicated sepsis in young infants and newborns hospitalized in the neonatal intensive care unit of a Taiwan hospital, during the period 2003-2020. Some molecular characteristics of the isolated Streptococcus agalactiae strains were also compared by serotyping, multilocus sequence typing and antibiotic susceptibility testing.

The message given by the paper is nevertheless difficult to find as the authors incorporated all of their analysis in the paper. The paper has to be strongly reduced and be focused only on what is original and important after the analysis of the data.

Specific comment

  1. The paper is already paginated as it will  be published. This is not a good procedure. It is difficult to make remarks as lines are not numerated.
  2. Please avoid the use of acronym throughout the paper.
  3. Title. Please use the international nomenclature for bacteria, especially in the title. Streptococcus agalactiae and not Group B streptococcus
  4. The Short title is not relevant for this paper (gram-negative bacteria).
  5. Line 17 of the Abstract. Type III/ST-17 represents a group of strains with diverse genetic backgrounds, it is not a strain.
  6. page 4. “The MLST analyses revealed” :

- “6 serotypes …” but 7 serotypes are depicted on figure 1A.

-“ 11 ST” but only 9  are present on figure 1B.

  1. Figures 1A, 1B, 1C; Fig. 2, Fig. 3 and Fig 4.

These figures are a mix of figures and tables. This is not a good way of proceeding. The tables should be suppressed from all these figures.

  1. Fig 1A,1B, 1C

Frequency of some strains is more than 100%. This is not possible.

  1. All figures. Add a legend on the X axis.
  2. Tables are not very clearly written.

Table 1. Please add a title for the data of the first column. For ‘gestional age’ and ‘birth body weight’, the number between parenthesis are not defined.

11. Line 8 of the Discussion. Please cite the table or figure were it is possible to see that ‘serotype Ib/CC12 GBS isolates were significantly more likely to cause complicated GBS sepsis’

12. Other more discriminating molecular methods could have been used to compare the bacterial strains

Author Response

RE: microorganisms-1384247

Complicated Group B Streptococcus sepsis with/without meningitis in young infants and newborns: the clinical and molecular characteristics and outcomes

Dear Editor,

Thank you for your appreciated comments on our manuscript. We had the manuscript revised, all according to the reviewers’ and editor’s suggestions. We underline every change and highlight in red color on the revised manuscript. The replies for the reviewers’ criticisms are as followings. We hope this revised version can be acceptable.

Best regards,

Ming-Horng Tsai

Chief, Division of Neonatology and Pediatric Hematology/Oncology, Department of Pediatrics, Yunlin Chang Gung Memorial Hospital, Taiwan, R.O.C.

Comments from Reviewer No.2:

Reviewer No.2

In this paper, Lin and collaborators compared cases of complicated and uncomplicated sepsis in young infants and newborns hospitalized in the neonatal intensive care unit of a Taiwan hospital, during the period 2003-2020. Some molecular characteristics of the isolated Streptococcus agalactiae strains were also compared by serotyping, multilocus sequence typing and antibiotic susceptibility testing.

The message given by the paper is nevertheless difficult to find as the authors incorporated all of their analysis in the paper. The paper has to be strongly reduced and be focused only on what is original and important after the analysis of the data.

Reply:

       Thank you for your instructive advice. I will strongly reduce the paper and focus only on what is original and important after analyzing the data.

Specific comment

  1. The paper is already paginated as it will be published. This is not a good procedure. It is difficult to make remarks as lines are not numerated.

Reply:

      Thank you for your instructive advice. It is the routine procedure of all the journals in the MDPI office. The good thing is to make a correction directly on the formatted document. Don’t worry about the lines. In the revised manuscript, I add the line numbers in the word file of the document, thank you.

  1. Please avoid the use of acronym throughout the paper.

Reply:

      Thank you for your instructive advice. I will avoid the use of acronym as much as I can throughout the paper, thank you. However, for GBS, ST, CC, MLST, EOS, and LOS, please allow me. In addition, I have listed all the abbreviations in the final part of the manuscript, thank you.

  1. Please use the international nomenclature for bacteria, especially in the title. Streptococcus agalactiaeand not Group B streptococcus

Reply:

      Thank you for your instructive advice.  I will use Streptococcus agalactiae to replace Group B streptococcus in the title, and use the international nomenclature for bacteria accordingly, thank you.

  1. The Short title is not relevant for this paper (gram-negative bacteria).

Reply:

Thank you for your instructive advice.  I will remove the short title accordingly, thank you.

  1. Line 17 of the Abstract. Type III/ST-17 represents a group of strains with diverse genetic backgrounds, it is not a strain

Reply:

      Thank you for your instructive advice. I revise it as type III/ST-17 isolates, not a strain, thank you.

  1. page 4. “The MLST analyses revealed”: “6 serotypes …” but 7 serotypes are depicted on figure 1A.-“ 11 ST” but only 9  are present on figure 1B.

Reply:

Thank you for your instructive advice.  I will revise it as 7 serotypes in line 15 of page 4. For 11 STs, because there are very few case numbers in 3 STs, there are grouped together as “others” in figure 1B. Therefore, only 9 are presented on figure 1B.

  1. Figures 1A, 1B, 1C; Fig. 2, Fig. 3 and Fig 4. These figures are a mix of figures and tables. This is not a good way of proceeding. The tables should be suppressed from all these figures.

Reply:

      Thank you for your instructive advice. In fact, all tables under the figures are to present the case number and percentage of all items. If the tables are suppressed from all these figures, there will be too many tables and figures. I beg your kindness to let the tables under the figures, thank you.

  1. Fig 1A,1B, 1C Frequency of some strains is more than 100%. This is not possible.

Reply:

      Thank you for your instructive advice. The frequencies of figure 1 are calculated by different subgroups. Because we categorized neonates with GBS bacteremia into two subgroups: complicated GBS sepsis (including cases of severe sepsis, septic shock and cases of GBS meningitis) and uncomplicated GBS bacteremia, the frequencies are calculated by the serotype (figure 1A), sequence types (figure 1B) and CC types (figure 1C) in these two subcategories. For example, in figure 1A, the white column of type Ia represents that type Ia GBS isolates are 15.1% of uncomplicated GBS bacteremia and 18.8% of complicated GBS sepsis. Therefore, there are some columns (especially type III, ST17 or CC17) will be more than 100% by sum. There are previous papers using these presentations. Please try to understand, thank you.

  1. All figures. Add a legend on the X axis.

Reply:

Thank you for your instructive advice. I will add a legend on the X axis, thank you. Please see the revised manuscript, the legend on the X axis of all figures, thank you.

  1. Tables are not very clearly written. Table 1. Please add a title for the data of the first column. For ‘gestational age’ and ‘birth body weight’, the number between parenthesis are not defined.

Reply:

Thank you for your instructive advice. I will add a title (Demographics) for the data of the first column. I will define the number between parenthesis as median (IQR), thank you.

  1. Line 8 of the Discussion. Please cite the table or figure were it is possible to see that ‘serotype Ib/CC12 GBS isolates were significantly more likely to cause complicated GBS sepsis

Reply:

Thank you for your instructive advice. I will cite the figure 1, which reveals that “serotype Ib/CC12 GBS isolates were significantly more likely to cause complicated GBS sepsis.

  1. Other more discriminating molecular methods could have been used to compare the bacterial strains

Reply:

       Thank you for your instructive advice. Sure there is other more discriminating molecular methods that could have been used to compare the bacterial strains. However, in addition to the molecular epidemiology, we aim to investigate the clinical impacts of complicated GBS sepsis, as well as the neurological sequela.